# In Vitro Neurotrophic Properties and Structural Characterization of a New Polysaccharide LTC-1 from *Pyrola corbieri* Levl (Luticao)

**DOI:** 10.3390/molecules28041544

**Published:** 2023-02-06

**Authors:** Liangqun Li, Kangkang Yu, Zhengchang Mo, Keling Yang, Fuxue Chen, Juan Yang

**Affiliations:** 1State Key Laboratory for Functions and Applications of Medicinal Plants, Guizhou Medical University, Guiyang 550014, China; 2The Key Laboratory of Chemistry for Natural Products of Guizhou Province and Chinese Academy of Sciences, Guiyang 550014, China; 3School of Life Science, Shanghai University, Shanghai 200444, China; 4Tongren Polytechnic College, Tongren 554300, China

**Keywords:** *Pyrola corbieri* Levl, polysaccharide, purification, structural analysis, neurotrophic activity

## Abstract

*Pyrola corbieri* Levl has been used to strengthen bones and nourish the kidney (the kidney governs the bone and is beneficial to the brain) by the local Miao people in China. However, the functional components and neurotrophic activity have not been reported. A new acidic homogeneous heteropolysaccharide named LTC-1 was obtained and characterized by periodate oxidation, Smith degradation, partial acid hydrolysis, GC–MS spectrometry, methylation analysis, and Fourier transform infrared spectroscopy, and its molecular weight was 3239 Da. The content of mannuronic acid (Man A) in LTC-1 was 46%, and the neutral sugar was composed of L-rhamnose (L-Rha), L-arabinose (L-Ara), D-xylose (D-Xyl), D-mannose (D-Man), D-glucose (D-Glc) and D-galactose (D-Gal) with a molar ratio of 1.00:3.63:0.86:1.30:6.97:1.30. The main chain of LTC-1 was composed of Glc, Gal, Man, Man A and the branched chain Ara, Glc, Gal. The terminal residues were composed of Glc and Gal. The main chain and branched chains were linked by (1→5)-linked-Ara, (1→3)-linked-Glc, (1→4)-linked-Glc, (1→6)-linked-Glc, (1→3)-linked-Gal, (1→6)-linked-Gal, (1→3, 6)-linked-Man and ManA. Meanwhile, neurotrophic activity was evaluated through PC12 and primary hippocampal neuronal cell models. LTC-1 exhibited neurotrophic activity in a concentration-dependent manner, which significantly induced the differentiation of PC12 cells, promoted the neurite outgrowth of PC12 cells, enhanced the formation of the web architecture of dendrites, and increased the density of dendritic spines in hippocampal neurons and the expression of PSD-95. These results displayed significant neurotrophic factor-like activity of LTC-1, which suggests that LTC-1 is a potential treatment option for neurodegenerative diseases.

## 1. Introduction

The progressive loss of neuronal cells is a common pathological symptom of neurodegenerative diseases in the brain [1]. It might be important to employ the localized delivery of neurotrophic factors to prevent or treat this cell loss [2]. Neurotrophin is an especially important protein that encourages the growth and differentiation of new neurons and synapses, maintains the function of mature neurons and partially protects neurons against cell death [3]. The beneficial effects of neurotrophic factors on depression and antidepressant diseases have been recognized previously [4,5,6]. However, protein neurotrophic factors have difficulty crossing the blood–brain barrier, resulting in difficulty in clinical administration [7]. When degraded before arriving at the action sites, only trace amounts of this kind of protein are in effect. Recently, apart from some molecules [8,9], it was reported that polysaccharides have neurotrophic factor-like effects, and a variety of natural polysaccharides, including MAPs from *Opuntia Milpa* Alta [10], polysaccharides from *Rosa roxburghii* [11], GLPS from *Ganoderma lucidum* and fucoid from brown seaweeds, were reported to exhibit neurotrophic or neuroprotective activity [12]. The neuroprotective effect of polysaccharides has received increasing attention.

*Pyrola* Linn. is a morphologically distinctive genus belonging to the Pyrolaceae family, according to the Angiosperm Phylogeny Group’s most recent classification [13]. This genus mostly occurs in the northern temperate zone, containing approximately 30 species, with 27 species in China. *P. corbieri* Levl. is predominantly distributed in karst areas in Guizhou Province, southwestern China. It has been used for the treatment of weakness, cough and kidney diseases for a long time in Chinese folk medicine and is used to nourish the kidney, strengthen bones, and enhance immunity by the local Miao people [14]. Modern pharmacological studies have demonstrated that *P. corbieri* Levl. herb has high nutritional and health value, good antioxidation, antibacterial and anti-inflammatory effects, and cartilage protection and can effectively improve the symptoms of coronary heart disease [15,16,17]. It has been reported that *P. corbieri* Levl. mainly contains polysaccharides, flavonoids, tannins, phenolic glycosides, quinones, etc. [18,19,20,21,22]. According to the theory of traditional Chinese medicine, the kidney governs bone and is beneficial to the medullary brain. To investigate the effect of invigorating the brain and neurotrophic activity of *P. corbieri* Levl, we obtained a new polysaccharide, LTC-1, and evaluated its neurotrophic activity through PC12 and primary hippocampal neuronal cell models.

## 2. Results

### 2.1. Extraction and Purification of LTC-1

The crude polysaccharide (158.5 g) was obtained by extracting the dried crushed whole herb of *P. corbieri* Levl (2.4 kg) by extraction with hot water and precipitation with ethanol. The resulting fraction was purified by DEAE-cellulose chromatography and a Sephacryl-300 column to give a purified polysaccharide, LTC-1. According to the phenolsulfuric acid assay and the *m*-hydroxydiphenyl method, the contents of neutral sugar and uronic acid in LTC-1 were 45% and 46%, respectively.

### 2.2. Physicochemical Properties of LTC-1

LTC-1 appeared as a white powder with [α] ^17.4^_D_ = +10° (C 0.1, H_2_O). The polysaccharide did not contain protein or nucleic acids according to the lack of absorption at 280 or 260 nm by UV spectrophotometry. GPC analysis of LTC-1 revealed a single symmetric peak, indicating that it was a homogeneous polysaccharide (shown in Appendix A). LTC-1’s average molecular weight was 3239 Da, according to GPC analysis.

The FT-IR spectrum of LTC-1 (Figure 1) revealed strong bands at 3423.59 and 2923.75 cm^−1^, which were attributed to the O-H and C-H vibrations of the polysaccharide. The C=O stretching vibration caused the band at 1623.36 cm^−1^. The peaks at 1323.35, 1242.11, 1038.33 and 1023.48 cm^−1^ corresponded to the O-H variable angle vibrations of glycoid hydroxyl (C-O-H) and C-H variable angle vibrations. The characteristic absorption peak at 937.23 cm^−1^ was due to the C-O-C stretching vibration of the pyran ring, while the band at 884.23 cm^−1^ was attributed to the variable-angle vibration of the C–H of the pyran ring.

### 2.3. Structural Characterization of LTC-1

The composition of LTC-1 was determined from the alditol acetates resulting from TFA hydrolysis, reduction, and acetylation. The GC–MS total ion current (TIC) chromatogram is shown in Figure 2. Compared with the retention time of the standard monosaccharide, LTC-1 was found to be composed of Rha, Ara, Xyl, Man, Glc, and Gal, with the molar ratios of the six monosaccharides being 1.00:3.63:0.86:1.30:6.97:1.30.

Appendix A displays the periodate oxidation results. The resulting reduction, hydrolysis, and acetylation were used to obtain the Smith degradation product, which was analyzed by GC–MS. Appendix A shows that formic acid was generated, indicating that the polysaccharide molecule contains (1→6)-linked or (1→) glycosidic bonds. Periodic acid consumption was more than twice that of formic acid, indicating that LTC-1 contains 1→2, 1→2,6, 1→4, and 1→4,6 glycosidic bonds. In the whole sugar component, an average of 0.27 mol of formic acid can be released per mol of hexose residues, which means that there were three nonreducing ends or 1→6 glycosidic bonds in every 10 hexose residues.

To determine the structure of glycosidic linkages, Smith degradation was used to selectively degrade the polysaccharide. The test results of Smith degradation products consisted of glycerol, erythritol and monosaccharides (Appendix A), further indicating that there were 1→, 1→2, 1→6, 1→2,6 bonds, 1→4, 1→4,6 bonds and 1→3 glycosidic linkages in LTC-1. This outcome needs to further prove the abovementioned conclusions by *methylation analysis*.

The result of LTC-1 partial hydrolysis (Appendix A) showed that the main chain edge monosaccharides or main chain end monosaccharides were Man, Glc and Gal; branched chain monosaccharides were mainly Ara, Man, Glc and Gal; and the core main chain monosaccharides were Man, Glu and Gal.

Due to the high content of uronic acid in LTC-1, GC–MS cannot determine the type of uronic acid and its connection mode after the methylation of polysaccharides without reduction of uronic acid. Through the analysis of the methylation products of polysaccharides after the reduction of uronic acid, the type of uronic acid and its connection mode in glycosides can be determined according to the changes in each monosaccharide before and after reduction.

Through the methylation of LTC-1 (Table 1) and methylation before and after uronic acid reduction of LTC-1 (Table 2), data analysis software and comparison with library data, the main chain of LTC-1 was composed of Glc, Man, Gal, Man A and the branched chain Ara, Glc, Gal. The terminal residues were composed of Glc and Gal. The main chain and branched chains are linked by (1→5)-linked-Ara, (1→3)-linked-Glc, (1→4)-linked-Glc, (1→6)-linked-Glc, (1→3)-linked-Gal, (1→6)-linked-Gal, (1→3,6)–linked -Man and Man A.

### 2.4. Effect of LTC-1 on Neurite Outgrowth in PC12 Cells

By designing a series of concentration gradients, LTC-1 was observed to promote the differentiation of PC12 cells in a dose-dependent manner. The higher the concentration, the closer the cell morphology was to neuron-like cells and the more similar the cell morphology of the positive control group. Especially when the polysaccharide concentration reached 30 μg/mL, cells with obvious neuron-like morphology were observed (Figure 3).

LTC-1 promoted the extension of PC12 cell protrusions in a dose-dependent manner. With increasing polysaccharide concentration, the proportion of protruding cells gradually increased. When the LTC-1 concentration was 10 μg/mL, the proportion of protruding cells was 13.33%. When the concentration was increased to 50 μg/mL, the proportion of protruding cells increased by nearly three times, to 38.2% (Figure 4A). However, the time dependence of LTC-1 in promoting protrusion extension was not obvious. After LTC-1 acted on the cells for 48 h, the change in the ratio of protrusion cells was not obvious, and even a slight decrease in the ratio of protrusion cells was found from the statistical chart (Figure 4B).

### 2.5. Effect of LTC-1 on Acetylcholinesterase Activity in PC12 Cells

The enhancement of acetylcholinesterase activity contributes to the extension of cell axons and is usually used as a sign of the end of neuronal differentiation in neurons [23]. Measuring the change in acetylcholinesterase activity of PC12 cells treated with a gradient concentration of LTC-1 showed that polysaccharides increased cholinease activity in a dose-dependent manner. As the concentration of polysaccharides increased, the activity of cholinesterase in cells also increased (Figure 5).

### 2.6. Effect of LTC-1 on GAP-43 Protein Expression in PC12 Cells

To understand the effect of the polysaccharide LTC-1 on the differentiation of PC12 cells at the protein level, the expression of growth-related protein (GAP-43) was detected by Western blotting. GAP-43, a neuron-specific phosphoprotein, is an intrinsic determinant and preferred molecular marker of synaptic plasticity and is closely related to the growth of neuronal neurites. GAP-43 expression increased in the cells treated with polysaccharides in a dose-dependent manner, as shown in Figure 6.

### 2.7. Effect of LTC-1 on the Length, Number and Density of Dendritic Dpines in Hippocampal Neurons

After 7 days of culture in vitro, hippocampal neurons entered the fifth stage of neuron growth and development [24], which was when synapses and neural networks began to form. Most cells present a multilevel neuron morphology. One or both ends of the cell body emit protrusions of different lengths, and the dendrites extending from the cell body have fewer branches and shorter lengths. At this time, the medium was changed, and 1 μM LTC-1 was added. The blank control group was the growth medium of neuronal cells. After 24 h of incubation, observation and data statistics were performed.

Compared with the control group, the LTC-1 group promoted the length of the dendrites, and the network shape formed by the dendrites was more obvious and denser (Figure 7A). The statistical data graph measured by ImageJ software shows (Figure 7B) that the length of the dendrites of neurons treated with LTC-1 was noticeably different from that of the control group (*** *p* < 0.001).

For a single neuron, the number of neuronal processes increased after LTC-1 treatment; that is, the number of dendritic branches extending from the neuron cell body increased significantly. Except for several trunk dendrites extending in different directions, the number of secondary branches increased (Figure 8A). The sum of the number of primary dendrites and secondary dendrites was counted in the experiment. In comparison to the blank control group, the number of cell protrusions in the experimental group increased significantly (Figure 8B).

After 24 h of administration, the dendritic spine density on hippocampal neurons in the LTC-1 experimental group increased significantly (Figure 9A). The statistical histogram also showed that compared with the control group, there were significant differences in the LTC-1 experimental group. (Figure 9B). * *p* < 0.05.

### 2.8. Effect of LTC-1 on the Expression of the Synapse Protein PSD-95

The synapse protein PSD95 is a scaffold protein widely present on the postsynaptic membrane and a member of the membrane-associated guanylate kinase (MAGUK) family [25]. After drug treatment, PSD-95 protein expression increased significantly compared to that in the control group (Figure 10).

## 3. Discussion

There are several methods for determining polysaccharide monosaccharide composition, and gas chromatography–mass spectrometry (GC–MS) has long been the preferred method due to its high sensitivity, simple processing, and time savings [26]. Therefore, this study investigates the composition of monosaccharides in LTC-1 using a gas-mass spectrometry method. In our previous study [18], at that time, it was also the first report of polysaccharide LTC-II from the whole herb *Pyrola corbieri* Levl. LTC-II was discovered to have a molecular weight of 22,000 Da. and found to be arabinose, mannose, glucose, galactose in a molar ratio of 35.2:1.0:13.4:4.2. The main chain of LTC-II is composed of glucose, mannose, galactose and the branched chain possessed arabinose, glucose, galactose. The main chain and branched chains are linked by (1→6)-linkaged glucose and (1→5)-linked-arabinose, respectively. In this study, a new acidic homogeneous heteropolysaccharide named LTC-1 differs significantly from LTC-II in terms of molecular weight, monomer composition, and content.

Neurological diseases are common neurological disorders that significantly affect patient quality of life. Neurotrophic factors (NTFS) have therapeutic implications for neurodegenerative disorders [27]. Therefore, it is important to conduct research on neurotrophins. In addition to proteins [28] and small molecules [29], a number of polysaccharides have also been reported to possess neuroprotective/neurotrophic effects in vitro and in vivo [30,31]. Neurons typically consist of a single axon and multiple dendrites, and neurite outgrowth is the first step for the formation of dendrites and axons, which is an important hallmark of neuron maturation and differentiation [32]. The elongation and branching of neuronal processes are critical links in the advancement of the brain and are key to the survival of the neuronal network system. Processes are of great significance for the construction of the entire neural circuit and the transmission of information between neurons [33]. The growth of neuronal processes is a key link in understanding and solving nerve regeneration and some degenerative diseases [34].

The hippocampus is one of the oldest neuromodulatory phylogenetic systems and has a highly polarized lamellar structure and a relatively independent distribution of neurons [35]. Due to their clear boundary with the surrounding brain tissue and easy separation, hippocampal neurons are often used in experiments as cell models for the study of neuroprotective properties [36]. At the same time, primary neuronal cells retain much of the activity of in situ neurons [37]. Therefore, it is more accurate to use it as a screening model for specific cell populations, closer to the specific location in vivo of cell origin, and suitable as an observation model for neurite outgrowth. 

In this study, LTC-1 neurotrophic activity was studied in combination with folk applications. Experiments on the neurotrophic activity of LTC-1 were carried out through PC12 and primary hippocampal neuronal cell models. These findings suggest that LTC-1 has neurotrophic factor-like activity. This exciting work not only enriches the types of neurotrophic active substances but also provides scientific data for the development and utilization of undeveloped natural resources.

## 4. Materials and Methods

### 4.1. Materials and Reagents

*Pyrola corbieri* Levl was collected in Guiyang city, Guizhou Province, PR China, in June 2018 and was authenticated by Prof. Qingwen Sun, Guizhou University of Traditional Chinese Medicine. A voucher specimen (No. 201806LTC) was deposited at the Key Laboratory of Chemistry for Natural Products of Guizhou Province and Chinese Academy of Sciences. DEAE-cellulose was supplied by Shanghai Hengxin Chemical Reagent Co., Ltd. (Shanghai, China), and Sephacryl-300 was purchased from Amersham Biosciences (Uppsala, Sweden). Galacturonic acid and the standards dextrans T-40 (Mw: 40 kDa), T-25 (Mw: 25 kDa), T-12 (Mw: 12 kDa), T-5 (Mw: 5 kDa) and T-1 (Mw: 1 kDa) were provided by Sigma Chemical Co. (MO, USA). D-mannose, D-glucose, D-galactose, D-xylose, D-fucose, L-rhamnose and L-arabinose were purchased from Shanghai Chemical Reagents Company (Shanghai, China). The pheochromocytoma cell line (PC12) was purchased from Shanghai Institute of Biochemistry and Cell Biology (Shanghai, China). The acetylcholinesterase kit and Pierce BCA protein kit were purchased from Shanghai Genmed Scientifics Co., Ltd. (Shanghai, China). GAP-43 (7B10) antibody was obtained from Santa Cruz Biotechnology, Inc. (Dallas, TX, USA). An Enhanced ECL Chemiluminescent Substrate Kit was purchased from PerkinElmer Life Sciences (Boston, MA, USA). Green fluorescent protein (GFP) transgenic mice (18 days gestation) were a kind gift from Professor Zhenggang Yang (Institute of Brain Science, Fudan University, Shanghai, China). Trypsin was acquired from Gibco-Life Technologies (Grand Island, NY, USA). DNAase and Trolox were purchased from Sangon Biotech Co., Ltd. (Shanghai, China).

### 4.2. General Analysis Methods

Ultraviolet–visible (UV–vis) spectra were acquired according to a Hewlett-Packard 8453 UV–vis spectrophotometer. Optical rotations were measured with an A22109 Autopol Manual Revision C Rudolph polarimeter (Rudolph, Hackettstown, NJ, USA). Fourier transform infrared (FT-IR) spectra of KBr pellets were obtained using a Bruker Vector-22 spectrometer. Using the phenolsulfuric acid method, the total carbohydrate content was calculated as D-glucose equivalents [38]. Uronic acid content was determined using an *m*-hydroxydiphenyl colorimetric method that does not interfere with neutral sugars [39].

### 4.3. Extraction and Purification of Polysaccharides

Using our previous literature as a guide, we extracted polysaccharides from LTC [18]. Dried crushed whole *Pyrola corbieri* Levl (4.5 kg) was refluxed three times for two hours with 80% ethanol (22 L) to remove some potential impurities, such as monosaccharides, oligosaccharides and fat-soluble substances. After alcohol extraction, the residues were extracted with distilled water (30 L) and refluxed every 2 h three times. The aqueous extract was vacuum-concentrated and treated with four volumes of ethanol overnight. The resulting precipitate was collected via filtration, washed with anhydrous ethanol and acetone, and then vacuum-dried. The precipitates were dissolved in ultrapure water and deproteinized by the Sevag method. After resolving the crude polysaccharide in distilled water, it was precipitated with anhydrous ethanol. The resulting precipitate (named LTC) was vacuum-dried at 60 ℃ and yielded a grayish power.

The LTC was eluted with distilled water using a DEAE-cellulose column (4.5 × 30 cm), and then a gradient of NaCl solution (0.05, 0.1, 0.2, 0.5, 1.0 M, respectively). The samples were further purified and eluted with 0.2 M NaCl on a Sephacryl-300 column (2.6 × 100 cm). The main fraction was gathered, concentrated, dialyzed, and lyophilized to produce LTC-1, a purified polysaccharide sample.

### 4.4. Structural Characterization of Polysaccharides [40]

#### 4.4.1. Homogeneity and Molecular Weight Determination 

The homogeneity and molecular weight of LTC-1 were evaluated by gel permeation chromatography (GPC) on an Agilent PL aquagel-OH 30 column (7.5 mm × 300 mm, 8 μm). LTC-1 was eluted with 0.05 M Na_2_SO_4_ at 1.0 mL/min and detected using a refractive-index detector at 35 °C. The TLC-1 molecular weight was determined by comparing it to a calibration curve created with different molecular weight dextran standards (1000, 5000, 12,000, 25,000, 40,000 Da) and glucose. The Workstation software package was used to collect and analyse all of the data provided by the GPC system.

#### 4.4.2. Monosaccharide Composition Analysis

Approximately 5 mg of LTC-1 was dissolved in trifluoroacetic acid (TFA, 2 M, 2 mL) and hydrolyzed at 100 °C for 8 h. After TFA was removed by vacuum evaporation, the precipitate was reduced in 2 mL distilled water with NaBH_4_ and acetylated with acetic anhydride (1 mL) and pyridine (1 mL). Gas chromatography–mass spectrometry (GC–MS) was used to analyse the resulting alditol acetates using an HP6890 instrument outfitted with an Agilent 19091S-433 HP-5MS quartz capillary column (30 m, 0.25 mm, 0.25 m) and an Agilent 5975C MS detector. The temperature program ranged from 50 °C to 300 °C at an 8 °C/min rate, with a 10-min stop at 300 °C.

#### 4.4.3. Periodate Oxidation-Smith Degradation

LTC-1 (25 mg) was oxidized in the presence of 15 mM NaIO_4_ (25 mL) and stored at 4 °C in the dark. The retest lasted an average interval of 5 h. NaIO_4_ consumption was quantitatively measured using the UV spectrophotometric method. Ethylene glycol was added to remove excess periodate after the sample solution was completely oxidized with stable absorbance. The periodate product solution (2 mL) was sampled to calculate the yield of formic acid by titration with 5 mM NaOH (bromocresol purple as an indicator), and the remainder was extensively dialyzed against distilled water for 48 h, vacuum concentrated below 40 ℃ and then reduced with NaBH_4_ for 24 h at room temperature and dialyzed again for 48 h. After the pH was adjusted to 5.5 using acetic acid, the mixture was dialyzed for 48 h against water. The alcohol derivative of polysaccharide was lyophilized and then hydrolyzed in 2 mL 2 M TFA at 120 ℃ for 2 h. After TFA was removed by vacuum evaporation, 1 mL acetic anhydride and 1 mL pyridine were used to acetylate the hydrolysate. The standard monosaccharides were converted into their respective acetates as described above and analyzed by GC–MS.

#### 4.4.4. Partial Acid Hydrolysis

LTC-1 (20 mg) was dissolved in 1 mL of 0.05 M TFA, hydrolyzed at 90 °C for 8 h, and then centrifuged at 4000 rpm for 10 min. The residue was marked as part I. After TFA evaporated from the supernatant, the hydrolysate was dialyzed for 24 h with distilled water, and then the outer part of the dialysis membrane was concentrated, dried and marked as part IV. The inner part was concentrated to a small volume, and anhydrous ethanol was added for alcohol precipitation to ensure that the alcohol concentration was not less than 80%, which was left overnight and centrifuged for 10 min at 4000 rpm. The precipitate was part II, and the supernatant was part III. Parts I–IV were operated according to the above “monosaccharide composition analysis” to prepare alditol acetate derivatives. The monosaccharide composition was determined by GC–MS.

#### 4.4.5. Methylation Analysis

LTC-1 (10 mg) was methylated three times in dimethyl sulfoxide with powdered sodium hydroxide and methyl iodide using the modified Ciucanu method described by Needs and Swlvendran [41]. The lack of hydroxyl absorption in the IR spectrum at 3400 cm^−1^ confirmed the completeness of methylation.

The permethylated product was depolymerized with 1 mL 85% formic acid at 100 °C for 4 h before being hydrolyzed with 2 M TFA for 6 h at 100 °C. The residues were reduced and acetylated. The resulting products were subjected to GC–MS analysis. The analysis and detection conditions were the same as above for the “monosaccharide composition analysis”.

LTC-1 (5 mg) was dissolved in 1 mL distilled water, 10 mg carbondimine was added, and the pH of the sample solution was adjusted to 4.8 using 0.04 M HCl, and the sample solution was placed at room temperature for 1 h. After 30 mg NaHB_4_ was added and reacted at 50 °C for 1.5 h, 2 M acetic acid was added to quench the reaction. The sample was subjected to dialysis with tap water for 48 h and distilled water for 24 h. The M-hydroxybiphenyl method was used to detect whether the reduction was complete.

After uronic acid reduction as described by Taylor and Conrad [42], the reduction intermediate was methylated according to the above “methylation analysis”.

### 4.5. In Vitro Neurotrophic Properties [43]

#### 4.5.1. PC12 Cell Culture and Administration Treatment

PC12 cells were grown in DMEM supplemented with 10% fetal bovine serum (FBS) and 5% horse serum (HS) at a density of approximately 3 × 10^4^ cells/cm^2^ per 6-well plate. The cells were cultured for 12 h at 37 ℃ in a humidified environment of 95% air and 5% CO_2_. Different concentrations of LTC-1 (0, 10, 20, 30, 40, 50 mg/mL) were added and further incubated for 48 h, and culture medium (DMEM + 1% FBS and 2% HS) of PC12 and NGF (final concentration: 30 ng/mL) was used as a blank control and positive control, respectively.

#### 4.5.2. Morphological Observation and Neurite Growth Analysis of PC12 Cells

A phase contrast microscope was used to examine cell morphology (Olympus CKX41; Tokyo, Japan), and neurite length was quantified. Cells were observed under a microscope with four to five visual fields randomly selected for each well, and images were captured. 3–5 significantly differentiated cells were chosen from each field, and the length of their longest neurite was measured. For each concentration, at least 100 cells were scored. Each data point was repeated three times.

#### 4.5.3. Measurement of Acetylcholinesterase Activity

PBS was added to the cells, mixed well and centrifuged for 5 min at 4 °C, and the supernatant was discarded. Cell lysate was added to the pellet, incubated in an ice bath for 30 min, and centrifuged for 5 min at 4 °C, and the supernatant was extracted into a 1.5 mL centrifuge tube. Ten microliters of supernatant was extracted for protein quantitative detection. The remaining supernatant was added to acetylthiocholine iodide and Ellman reagent to react and then immediately placed into a microplate reader, and the readings were read at 0 and 15 min at 412 nm wavelength.

#### 4.5.4. GFP Transgenic Mouse Hippocampal Neuron Culture and Administration Treatment

Hippocampi were extracted from 18-day-old GFP-transgenic mouse embryos, and neuronal cell culture was carried out as previously described. Briefly, hippocampi were placed in an ice-cold balanced salt solution in a sterile 35 mm petri dish. The hippocampal tissues were isolated and chopped into three pieces of 1 mm under a microscissor and digested with 0.25% trypsin, HBSS and 1 μg/mL DNAase at 37 ℃ for 15 min. The enzyme reaction was terminated by adding 1 mL FBS. One-third of the medium was changed every two days, and the cells were cultured up to day 14. Polysaccharide and 0.1 mM Trolox were added to the medium.

#### 4.5.5. Long-Term Recording of Living Cells

An X-Y automated positioning system was used to resolve cell positions over multiple time points. The primary and secondary dendrites with clear dendritic spines were selected for experimental recording, and photographs were taken every 0.5 h and continued for 24 h at Living Cell Station (PerkinElmer, Boston, MA, USA).

The dendrite length was calculated from the neuron cell body to the dendrite terminal. When there was a branch, one of the branches was included in the dendrite of the previous level, and the length of at least 30 dendrites was counted; then, the average value was obtained.

Averaged from >300 dendritic spines out of 10–30 dendrites on at least 6 neurons. The length of a dendritic spine was the distance from the root on the dendrite to the very tip of the head, and the size of the head depended on the width at the widest point of the head of the spine. For the measurement of dendritic spine size, a clear area 50–200 μm from the soma in the primary dendritic spines of each neuron was used for statistics.

Its density statistics include all dendritic spines on the entire dendritic section. The size of all dendritic spines on the neuron was counted first, and the average value of each neuron was taken. Then, the average value of all neurons was taken.

#### 4.5.6. Western Blot Analysis

Protein concentrations in total cell lysates were determined using a Pierce BCA protein kit and the manufacturer’s instructions. The cultures were rinsed three times with D-Hanks before being scraped and homogenized in ice-cold lysis buffer and centrifuged for supernatant collection. Each lane of an 8% sodium dodecyl sulfate polyacrylamide gel (SDS–PAGE) was loaded with approximately 15 g for electrophoresis and electrophoretically transferred to polyvinylidene fluoride (PVDF) membranes using standard procedures. The membranes were blocked at room temperature with 5% nonfat dry milk in Tris-buffered saline with Tween (TBST; 20 mM Tris-HCl, 137 mM NaCl and 0.1% Tween-20; pH 7.6) for 2 h and then incubated at 4 °C overnight with the primary antibody against PSD-95 (1:2000 dilution). The blots were washed three times in TBST for 10 min each and then incubated for 2 h with horseradish peroxidase (HRP)-conjugated donkey anti-mouse IgG. An enhanced chemiluminescent (ECL) substrate kit was used to detect immunoreactivity signals. The gray density level difference was determined using Photoshop software. As an internal control, a rabbit anti-GAPDH monoclonal antibody was used.

### 4.6. Statistical Analysis

In accordance with Dhutia et al. (2021) [44], using the Statistical Package for Social Sciences 18.0 program, the data were expressed as the mean standard error of the mean (SEM) (SPSS Inc., Chicago, IL, USA). The Tukey–Kramer test was used to assess significant differences after one-way analysis of variance (ANOVA). *p* < 0.05 was considered statistically significant.

## 5. Conclusions

In our present study, a new polysaccharide named LTC-1 was isolated from the whole herb *Pyrola corbieri* Levl, which has been used for the treatment of strengthening to strengthen bones and nourish the kidney by the local indigenous people. However, LTC-1 is urgently needed to confirm its molecular structure through NMR data, it was not performed because a solvent that would dissolve it well could not be found. Meanwhile, LTC-1 demonstrated significant neurotrophic factor-like activity, increasing neurite length, enhanced the formation of the web architecture of dendrites and increased the density of dendritic spines in vitro. We will conduct in vivo experiments of neurotrophic properties in the future. Therefore, LTC-1 has tremendous potential as a brain-protective agent or as a potential treatment for neurodegenerative diseases.

## Figures and Tables

**Figure 1 molecules-28-01544-f001:**
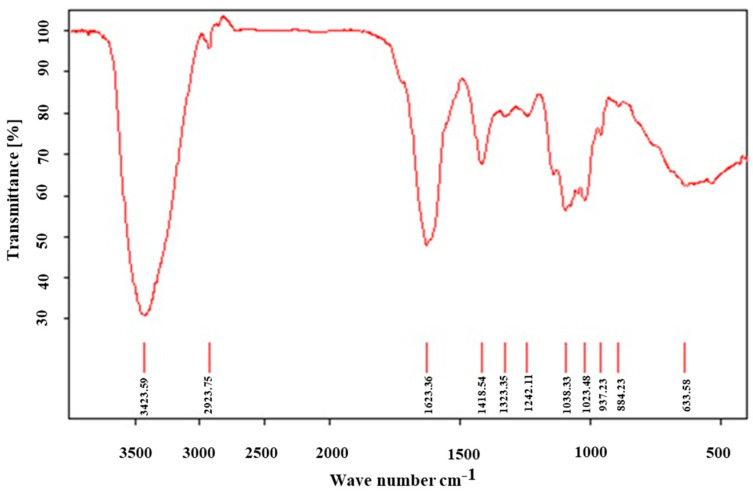
FR-IR spectrum of LTC-1 from *P. corbieri* Levl.

**Figure 2 molecules-28-01544-f002:**
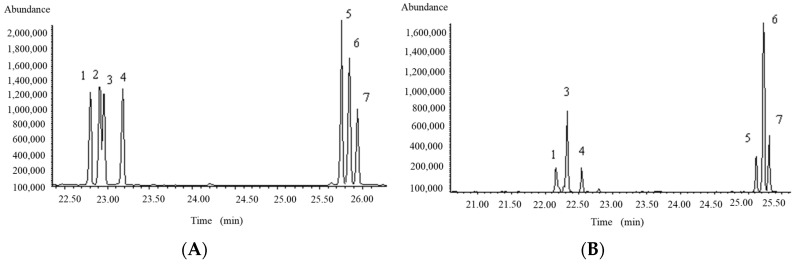
GC–MS TIC of standard monosaccharides (**A**) and monosaccharide components of LTC-1 (**B**). 1. Rha 2. Fuc 3. Ara 4. Xyl 5. Man 6. Glc 7. Gal.

**Figure 3 molecules-28-01544-f003:**
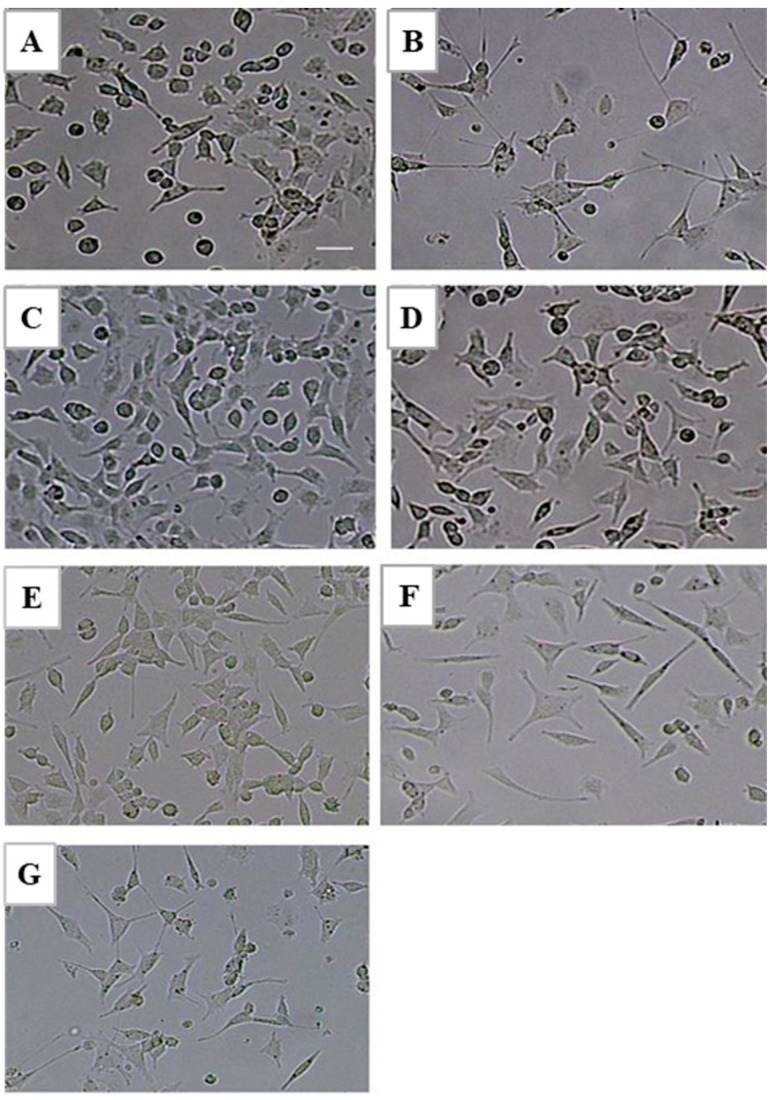
A gradient concentration of LTC-1 promotes the differentiation of PC12 cells. (**A**) Blank control group; (**B**) 30 ng/mL NGF group; (**C**–**G**) gradient concentration of LTC-1 groups: (**C**) 10 μg/mL; (**D**) 20 μg/mL; (**E**) 30 μg/mL; (**F**) 40 μg/mL; (**G**) 50 μg/mL.

**Figure 4 molecules-28-01544-f004:**
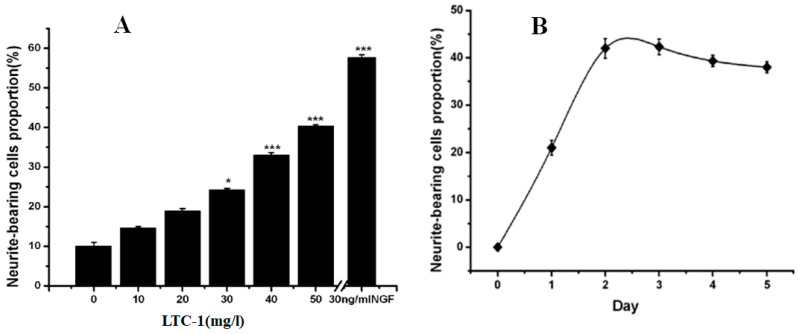
LTC-1 promotes the growth of PC12 cell neurites. (**A**) The relationship between LTC-1 promotion of PC12 cell neurite growth and concentration; (**B**) The relationship between LTC-1 promotion of PC12 cell neurite growth and time. The control group was defined as the group that received only NGF (1 mg/L). The neurite lengths of PC12 cells were measured after 4 days. The data are presented as the mean ± SD (n = 100). One-way ANOVA was used for statistical analysis, followed by Tukey’s multiple comparison test, * *p* < 0.05, *** *p* < 0.001 vs. control.

**Figure 5 molecules-28-01544-f005:**
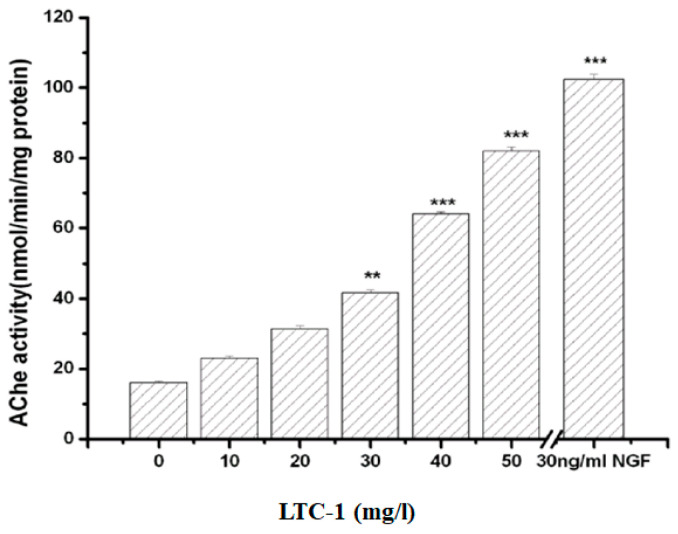
LTC-1 dose-dependently enhances the activity of acetylcholinesterase in cells. The mean SD is used to express the data. One-way ANOVA was used for statistical analysis, followed by Tukey’s multiple comparison test, ** *p* < 0.01, *** *p* < 0.001 vs. control.

**Figure 6 molecules-28-01544-f006:**
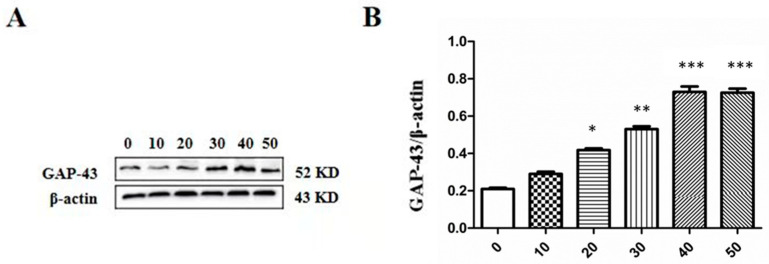
Effect of LTC-1 on the protein expression of GAP-43 in PC12 cells. (**A**) Proteins in PC12 cells were analyzed by Western blot analysis. (**B**) The relative optical density was normalized to *β*-actin. Quantitation of GAP-43 protein levels in the PC12 cells of LTC-1 six independent experiments. The data are presented as the mean ± SD (n = 3). One-way ANOVA was used for statistical analysis, followed by Tukey’s multiple comparison test, * *p* < 0.05, ** *p* < 0.01, *** *p* < 0.001 vs. control.

**Figure 7 molecules-28-01544-f007:**
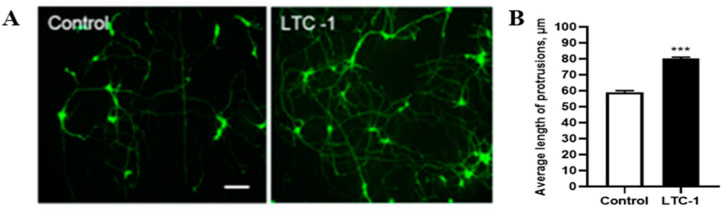
The length of dendrites of hippocampal neurons increased after treatment with LTC1. (**A**) The growth status of hippocampal neuron dendrites 24 h after drug treatment. Ruler = 20 μm; (**B**) statistical graph of dendritic length 24 h after administration. The mean (SD) is used to express the data. One-way ANOVA was used for statistical analysis, followed by Tukey’s multiple comparison test, *** *p* < 0.001 vs. control.

**Figure 8 molecules-28-01544-f008:**
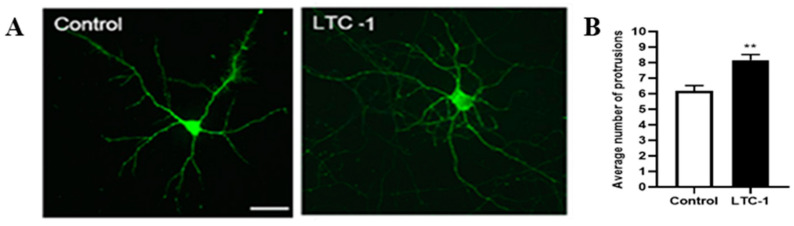
The number of dendrites of hippocampal neurons increased after treatment with LTC-1. (**A**) The growth status of single hippocampal neuron dendrites 24 h after drug treatment. (**B**) Statistical graph of the sum of the number of primary and secondary dendrites after administration. The mean (SD) is used to express the data. One-way ANOVA was used for statistical analysis, followed by Tukey’s multiple comparison test, ** *p* < 0.01 vs. the control group.

**Figure 9 molecules-28-01544-f009:**
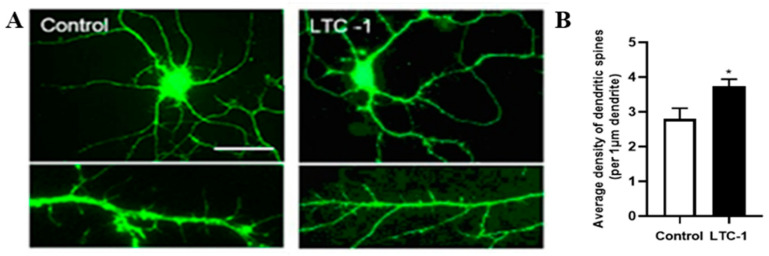
The dendritic spine density of hippocampal neurons increased after treatment with LTC-1. (**A**) The growth status of hippocampal neuron dendritic spines after 24 h of treatment. Scale = 10 μm. (**B**) Statistical graph of dendritic spine density. The mean and standard deviation are used to express the data. One-way ANOVA was used in the statistical analysis, followed by Tukey’s multiple comparison test, * *p* < 0.05 vs. control group.

**Figure 10 molecules-28-01544-f010:**
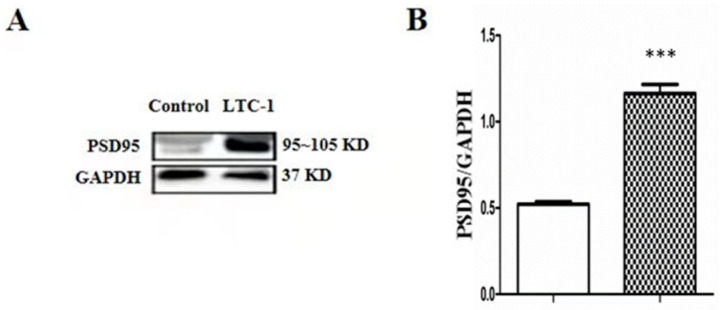
Effect of LTC-1 on the protein expression of PSD95 in hippocampal neurons. (**A**) PSD95 Western blotting results in hippocampal neurons. (**B**) The relative optical density was normalized to GAPDH. One-way ANOVA was used for statistical analysis, followed by Tukey’s multiple comparison test, *** *p* < 0.001 vs. control.

**Table 1 molecules-28-01544-t001:** GC–MS results of methylated LTC-1.

Retention Time (Min)	Methylation Position	Linked Type	Molar Ratio	Characteristics of Fragment Ions
19.10	2,3-Ara	1→5	2.00	101, 117, 161
20.52	2,4,6-Glu	1→3	2.05	87, 101, 117, 129, 161, 189
21.05	2,3,4,6-Glu	1→	11.20	71, 87, 101, 117, 129, 145, 161
21.34	2,3,4,6-Gal	1→	4.27	71, 87, 101, 117, 129, 145, 161, 205
22.43	2,3,6-Glu	1→4	28.74	71, 87, 101, 117, 129, 233
22.53	2,4,6-Gal	1→3	1.00	71, 87, 101, 117, 129, 233
22.66	2,3,4-Glu	1→6	2.13	71, 87, 101, 117, 129, 161, 189, 233
23.05	2,3,4-Gal	1→6	1.01	71, 87, 101, 117, 129, 161, 189, 233
24.16	2,4-Man	1→3,6	1.90	58, 87, 129, 159, 173, 189

**Table 2 molecules-28-01544-t002:** GC–MS results of methylation before and after uronic acid reduction of LTC-1.

Methylation before Reduction					Methylated after Reduction				
Methylation Position	Linked Type	Molar Ratio	Total Moles	Ratio	Ratio	Total Moles	Molar Ratio	Linked Type	Methylation Position
2,3-Ara	1→5	1.78	1.78	4%	5%	1.59	1.59	1→5	2,3-Ara
2,3,4,6-Gal	1→	3.55	5.53	12%	10%	3.12	1.59	1→	2,3,4,6-Gal
2,3,4-Gal	1→6	0.98					0.53	1→6	2,3,4-Gal
2,4,6-Gal	1→3	1.00					1.00	1→3	2,4,6-Gal
2,3,4,6-Glc	1→	7.74	39.58	81%	75%	22.79	3.12	1→	2,3,4,6-Glc
2,3,6-Glc	1→4	27.49					18.23	1→4	2,3,6-Glc
2,3,4-Glc	1→6	1.58					1.44	1→6	2,3,4-Glc
2,4,6-Glc	1→3	2.77							
2,4-Man	1→3,6	0.94	0.94	3%	8%	2.60	2.60	1→5	2,4-Man

## Data Availability

Data for the compounds are available from the authors.

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
