# Peer review of "In Vitro Neurotrophic Properties and Structural Characterization of a New Polysaccharide LTC-1 from Pyrola corbieri Levl (Luticao)"

_molecules, 2023, doi:10.3390/molecules28041544_

Round 1
Reviewer 1 Report
The manuscript entitled In Vitro Neurotrophic Properties of a New Polysaccharide LTC- 21, an Essential Component of Pyrola Corbieri Levl (Luticao)" is well designed and nicely written. However, there are some minor concerns need to be improved:
1. The title is too long and can be shortened.
2. Authors should add more details and references to the introduction part. I suggest you including these refernces:
https://doi.org/10.3390/molecules28010306
https://doi.org/10.3390/ijms24020986
https://doi.org/10.3390/ijms24021329
3.The quality of figure 1 has to improved as the numbers are not clear.
4.Future work and authors opinion have to be included in the conclusions.
Author Response
Response to Reviewer Comments
Dear Reviewer,
On behalf of my co-authors, we thank you very much for brining us an opportunity to revise our manuscript again, we appreciate editor and reviewers very much for their positive and constructive comments and suggestions on our entitled “In Vitro Neurotrophic Properties of a New Polysaccharide LTC-1, an Essential Component of Pyrola Corbieri Levl (Luticao)” (molecules -2188049)
We have studied reviewer’s comments carefully and have made revison which marked in blue in the paper. We have tried our best to revise our manuscript according to the comments. Attached please find the revised version, which we would like to submit for your kind consideration. We would like to express our great appreciation to you and reviewers for comments on our paper. Looking forward to hearing from you.
Thank you and best regards.
Yours sincerely,
Juan Yang
E-mail: yangxz2002@126.com
Point 1: English language and style (x) Moderate English changes required.
Response 1: We have moderately and carefully improved and revised the english language and style of the entire manuscript (in red)
Point 2: Does the introduction provide sufficient background and include all relevant references? Must be improved
Response 2: We have added some background and include 4 relevant references in the 1. Introduction section (in red)
Point 3: Are all the cited references relevant to the research? Can be improved
Response 3: We have adapted and refined some of the literature in the References (in red)
Point 4: Are the results clearly presented? Can be improved
Response 4: We have refined the presentation of some results to be very clear in the 2.Results (in red)
Point 5: Are the conclusions supported by the results? Can be improved
Response 5: We have refined our conclusions by closely following the results (in red)
Point 6: Comments and Suggestions for Authors (1) The title is too long and can be shortened.
Response 6: We have changed the title to “In Vitro Neurotrophic Properties and Structural Characterization of a New Polysaccharide LTC-1 from Pyrola Corbieri Levl (Luticao) “(in red)
Point 7: Comments and Suggestions for Authors: (2) Authors should add more details and references to the introduction part. I suggest you including these refernces: https://doi.org/10.3390/molecules28010306
https://doi.org/10.3390/ijms24020986
https://doi.org/10.3390/ijms24021329
Response 7: We have added more details and include 3 relevant references listed by reviewer in the introduction part (in red)
Point 8: Comments and Suggestions for Authors: (3) The quality of figure 1 has to improved as the numbers are not clear
Response 8: We have replaced it with a clearly picture 1 (At the location of Fig. 1)
Point 9: Comments and Suggestions for Authors :(4) Future work and authors opinion have to be included in the conclusions
Response 9: We have refined our conclusions to incorporate future work and opinion in the 5.Conclusions part. (in red)

Reviewer 2 Report
The study named "In Vitro Neurotrophic Properties of a New Polysaccharide LTC-2, an Essential Component of Pyrola Corbieri Levl (Luticao)" is a complex study with both prepared cell cultures and the preparation of Polysaccharide LTC-1 from herbal Pyrola Corbieri Levl. the results clearly show that LTC-1 can significantly induce the differentiation of ( pheochromocytoma cell line)cells, ranging from 0 mg/l to 50 mg/l, in
a concentration-dependent manner. The in vitro biological activity assay, which was cultured with hippocampal neurons, revealed that LTC-1 increased neurite length, enhanced the formation of the web architecture of dendrites and increased the density of dendritic spines. Meanwhile, western blot analysis showed that after treatment with LTC-1, the expression of PSD-95 was significantly enhanced. These findings suggest that LTC-1 has neurotrophic factor-like activity.
Author Response
Response to Reviewer Comments
Dear Reviewer,
On behalf of my co-authors, we thank you very much for briving us an opportunity to revise our manuscript again, we appreciate editor and reviewers very much for their positive and constructive comments and suggestions on our entitled “In Vitro Neurotrophic Properties of a New Polysaccharide LTC-1, an Essential Component of Pyrola Corbieri Levl (Luticao)” (molecules -2188049)
We have studied reviewer’s comments carefully and have made revison which marked in blue in the paper. We have tried our best to revise our manuscript according to the comments. Attached please find the revised version, which we would like to submit for your kind consideration. We would like to express our great appreciation to you and reviewers for comments on our paper. Looking forward to hearing from you.
Thank you and best regards.
Yours sincerely,
Juan Yang
E-mail: yangxz2002@126.com
Point 1: English language and style (x)English language and style are fine/minor spell check required.
Response 1: We have moderately and carefully improved and revised the english language and style of the entire manuscript (in red)
Point 2: Comments and Suggestions for Authors
The study named "In Vitro Neurotrophic Properties of a New Polysaccharide LTC-2, an Essential Component of Pyrola Corbieri Levl (Luticao)" is a complex study with both prepared cell cultures and the preparation of Polysaccharide LTC-1 from herbal Pyrola Corbieri Levl. the results clearly show that LTC-1 can significantly induce the differentiation of ( pheochromocytoma cell line)cells, ranging from 0 mg/l to 50 mg/l, in a concentration-dependent manner. The in vitro biological activity assay, which was cultured with hippocampal neurons, revealed that LTC-1 increased neurite length, enhanced the formation of the web architecture of dendrites and increased the density of dendritic spines. Meanwhile, western blot analysis showed that after treatment with LTC-1, the expression of PSD-95 was significantly enhanced. These findings suggest that LTC-1 has neurotrophic factor-like activity
Response 2: We are grateful to the reviewer for your thorough summaries.

Reviewer 3 Report
GENERAL COMMENTS
This Manuscript aimed to extract a polysaccharide, LTC-1 from Pyrola Corbieri Levl. (Luticao), and evaluate the effect of invigorating the brain and neurotrophic activity, through PC12 and primary hippocampal neuronal cell models.
The idea, research work and the results represented are interesting in the field; however, the MS needs some revisions for enhanced form for publication.
The comments and questions provided below may be considered as a guide to the authors to put their work into more appropriate form for publication.
SPECIFIC COMMENTS
1. Discussion section is poorly written which failed to reflect the good quality conducted work, on the other hand some discussions are included within the Results section. I suggest that either the authors choose to merge the two sections to be complete, informative and do not make readers to be confused, or rewrite both sections to be more informative with needful comparisons with previous research work in the field.
2. Provide references to "all" described methods even Statistical analyses; as no references mentioned for any methodology except for Methylation Analysis! Revise thoroughly.
3. Revise to provide higher resolution of unclear provided figures, e.g. Fig. 1, 3.
4. Update references till the current year.
Wish you all the best,,
Author Response
Response to Reviewer Comments
Dear Reviewer,
On behalf of my co-authors, we thank you very much for briving us an opportunity to revise our manuscript again, we appreciate editor and reviewers very much for their positive and constructive comments and suggestions on our entitled “In Vitro Neurotrophic Properties of a New Polysaccharide LTC-1, an Essential Component of Pyrola Corbieri Levl (Luticao)” (molecules -2188049)
We have studied reviewer’s comments carefully and have made revison which marked in blue in the paper. We have tried our best to revise our manuscript according to the comments. Attached please find the revised version, which we would like to submit for your kind consideration. We would like to express our great appreciation to you and reviewers for comments on our paper. Looking forward to hearing from you.
Thank you and best regards.
Yours sincerely,
Juan Yang
E-mail: yangxz2002@126.com
Point 1: English language and style (x) English language and style are fine/minor spell check required.
Response 1: We have moderately and carefully improved and revised the english language and style of the entire manuscript (in red)
Point 2: Are all the cited references relevant to the research? Can be improved
Response 2: We have adapted and refined some of the literature in the References (in red)
Point 3: Is the research design appropriate? Must be improved
Response3: We have improved the research design of the manuscript in text (in red)
Point 4: Are the methods adequately described? Can be improved
Response 4: We have improved the description of each method in the manuscript (in red)
Point 5: Are the results clearly presented? Can be improved
Response 5: We have refined the presentation of some results to be very clear in the 2.Results (in red)
Point 6: Are the conclusions supported by the results? Can be improved
Response 6: We have refined our conclusions by closely following the results (in red)
Point 7: Comments and Suggestions for Authors (1) Discussion section is poorly written which failed to reflect the good quality conducted work, on the other hand some discussions are included within the Results section. I suggest that either the authors choose to merge the two sections to be complete, informative and do not make readers to be confused, or rewrite both sections to be more informative with needful comparisons with previous research work in the field.
Response 7: We have rewrite both sections to be more informative with needful comparisons with previous research work in the field. (in red)
Point 8: Comments and Suggestions for Authors: (2) Provide references to "all" described methods even Statistical analyses; as no references mentioned for any methodology except for Methylation Analysis! Revise thoroughly.
Response 8: We have provided references to "all" described methods even Statistical analyses thoroughly. (in red)
Point 9: Comments and Suggestions for Authors: (3) Revise to provide higher resolution of unclear provided figures, e.g. Fig. 1, 3.
Response 9: We have replaced them with a clearly picture 1 and 3 (At the location of Fig. 1 and Fig. 1 , respectively)
Point 10: Comments and Suggestions for Authors (4) Update references till the current year.
Response 10: We have updated references till the current year references part. (in red)
